# Big data driven perovskite solar cell stability analysis

Zhuang Zhang[1], Huanhuan Wang[1], T. Jesper Jacobsson[1] & Jingshan Luo [1,2] ✉

During the last decade lead halide perovskites have shown great potential for photovoltaic applications. However, the stability of perovskite solar cells still restricts commercialization, and lack of properly implemented unified stability testing and disseminating standards makes it difficult to compare historical stability data for evaluating promising routes towards better device stability. Here, we propose a single indicator to describe device stability that normalizes the stability results with respect to different environmental stress conditions which enables a direct comparison of different stability results. Based on this indicator and an open dataset of heterogeneous stability data of over 7000 devices, we have conducted a statistical analysis to assess the effect of different stability improvement strategies. This provides important insights for achieving more stable perovskite solar cells and we also provide suggestions for future directions in the perovskite solar cell field based on big data utilization.

The last decade has witnessed a rapid technological rush aimed at the development of emerging devices for solar energy conversion such as dye-sensitized cells[1], perovskite cells[2], and integrated devices[3]. Since the halide perovskites were introduced as visible-light sensitizers in 2009[4], perovskite solar cells (PSCs) have witnessed remarkable progress in terms of photoelectric conversion efficiency (PCE) with record certified PCE now reaching 25.7%[5], which is comparable with single crystal silicon solar cells. Device stability does, however, remain a problem, despite much progress.

There are many causes of PSC degradation, but some of the most common triggers are moisture[6], heat[7], and light[8] (Fig. 1a). The perovskite layer, as well as the charge transport layers, can decompose or undergo phase transitions, and all the interfaces between those layers are susceptible to undesired changes. Several strategies for improving PSC stability have been reported and, while they tend to be successful within the reported scope, they are hard to evaluate and compare due to a lack of consistently reported stability data and proper statistical analysis. Such a statistical analysis requires a large amount of stability data for various types of devices, which previously has been lacking, but now is available thanks to the Perovskite Database Project[9,10]. In that open database, device data for over

42,400 devices have been collected, where stability data are available in around 7500 cases. This makes the comparison of device parameters and macroscopic analysis of stability strategies possible. However, a remaining problem for statistical analysis of the stability data is that widely different measurement conditions and reporting standards have been used within the perovskite community over the last decade.

In this work, we have developed a set of heuristics that enable a rough comparison of stability data and consider different levels of stress in terms of heat, moisture, and illumination under the stability measurement. This has been used to perform a statistical analysis of all devices with stability data reported in the Perovskite Database with the aim to compare different strategies for increasing PCE stability related to perovskite composition, device architecture and the charge transport layers. Meaningful suggestions are also made for the future development of the PSC field.

## Results

### The indicator for stability analysis

Due to different stability testing conditions and data formats used in different papers, the available stability data are not directly

[1]Institute of Photoelectronic Thin Film Devices and Technology, Solar Energy Research Center, Key Laboratory of Photoelectronic Thin Film Devices and Technology of Tianjin, Ministry of Education Engineering Research Center of Thin Film Photoelectronic Technology, Renewable Energy Conversion and Storage Center, Nankai University, 300350 Tianjin, China. [2]Haihe Laboratory of Sustainable Chemical Transformations, 300192 Tianjin, China. ✉e-mail: jingshan.luo@nankai.edu.cn

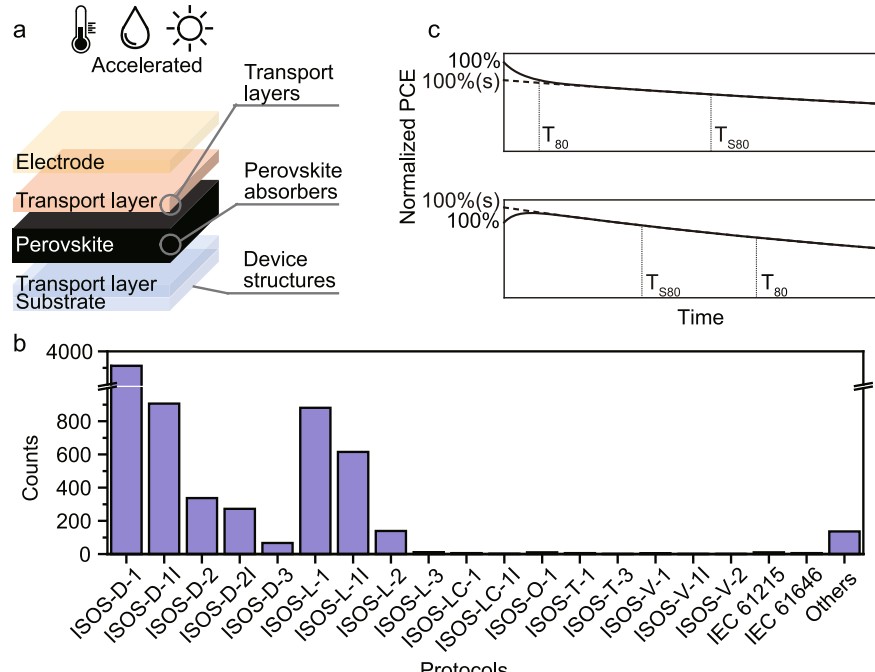

**Fig. 1 | Diagrams of perovskite solar cell stability tests. a** General device architecture of a perovskite solar cell. **b** The distribution of stability protocols used for stability data in the Perovskite Database. **c** Two possible efficiency decay curves of perovskite solar cells illustrating different types of burn-in behaviors followed by a slower exponential decay. (PCE stands for photoelectric conversion efficiency. $T_{80}$ is the time for PCE to decay to 80% of the initial efficiency. $T_{S80}$ is the time for PCE to decay to 80% of the stabilized efficiency (marked as 100%(s)).

comparable. That would require a unified indicator based on the protocols and consensus of photovoltaic research.

A reasonable starting point for constructing such an indicator is the consensus statement for stability assessment and reporting for perovskite photovoltaics based on the International Summit on Organic Photovoltaic Stability (ISOS) procedures proposed by Khenkin et al.[11], which is a development of stability protocols developed for the Organic Photovoltaic (OPV) community[12]. Those protocols include stability testing standards under different environmental stress, i.e., dark storage (ISOS-D), bias stability (ISOS-V), light soaking (ISOS-L), outdoor stability (ISOS-O), thermal cycling (ISOS-T), light cycling (ISOS-LC), solar-thermal cycling (ISOS-LT), each of which has three different levels determined by the level of thermal stress, humidity, light and circuit bias.

Stability data based on all of those various protocols are with different frequencies found in the Perovskite Database (Fig. 1b). As these protocols are associated with different environmental stresses, only devices measured with the same protocol can be directly compared. The most widely used protocol is ISOS-D-1, where devices are stored in ambient air in the dark at open circuit, i.e., stored in a drawer and periodically measured. The effect of environmental stress is not reflected in this protocol, thus making it non-ideal for directly assessing the operational device stability. However, other protocols are used less frequently and datasets with less than a few hundred data points prevent meaningful statistical results. If the data could be converted to the same environmental stress level, more useful data would thus be gained.

In some accelerated aging tests, the acceleration factor derived from a physics model is used to relate the times to failure under a high environmental stress condition, to a reference situation, making it possible to predict long-term device degradation. Here, we consider acceleration factors for several major environmental stresses to normalize the stability results under different test conditions.

One of the most important environmental stresses is temperature, and the effect of temperature has been discussed in the stability assessment consensuses of both OPVs and perovskite photovoltaics[11,12]. For temperature stress experiments, a simple Arrhenius model is widely used to describe the device performance decay rate, $k$, as a function of temperature[13], $T$.

$$k = A \cdot e^{\frac{-E_a}{k_B T}} \tag{1}$$

$E_a$ is the effective activation energy of the degradation process, $k_B$ is the Boltzmann constant, and $A$ is a proportionality constant. Previous reports have shown that the Arrhenius model can fit the degradation rate of perovskite films and PSCs reasonably well and that the effect of temperature is independent of other stress factors[14]. As the time to failure is inversely proportional to the degradation rate, the ratio of the degradation rates measured at two different temperatures corresponds to the ratio between the times it takes to decay to a specific amount of the initial performance, e.g., 80%. If room temperature (300 K) is considered as the reference condition, the decay rate at temperature $T$, $k(T)$, divided by the decay rate at 300 K (Eq. 2) can thus be interpreted as an acceleration factor, $A_{temperature}(T)$, which represents how much the time to failure will be extended when converting the experimental data to the reference condition.

$$A_{temperature}(T) = \frac{k(T)}{k(300K)} = e^{\frac{-E_a}{k_B}\left(\frac{1}{T}-\frac{1}{300}\right)} \tag{2}$$

Experimental reports on PSC stability have, based on temperature-dependent efficiency decay, estimated the effective activation energy, $E_a$, to be in the range 0.6–0.725 eV (0.634–0.725 eV[15] and 0.60–0.68 eV[7]), with some of the variation ascribed to different device structures. The temperature-dependent degradation of perovskite devices will most likely not follow a simple Arrhenius behavior with the same activation energy for all types of device configurations and material compositions. It could, however, provide a first estimate that is good enough to enable better comparison between historic stability data. Based on previous heuristics we have used an

approximation of 0.6 eV for $E_a$ for all stability data in this work. In practice, this assumes that a device that is stable for 1 h at 85 °C would be stable for around 43 h at room temperature.

Another widely studied environmental stress is humidity. According to previous reports, water molecules can penetrate the perovskite structure and form an intermediate phase of monohydrate perovskite[6,16]. Generally, for an elementary reaction, the reaction order in the reaction rate equation of each reactant is equal to its stoichiometric coefficient, so under the assumption of a rate-determining first-step reaction between perovskite and water, the consumption rate of perovskite would be approximately proportional to the relative humidity (RH). Recently, Timothy et al.[14] developed a kinetic model for the degradation rate of MAPbI$_3$ films, which states that the reaction rate of the dominant water-accelerated photooxidation process is proportional to the partial pressure of water vapor with a coefficient of $1.3 \pm 0.3$. However, under low moisture stress, dry photooxidation and thermal decomposition will dominate the degradation and the humidity seems to be largely irrelevant. This is equivalent to setting a threshold to cut off the low humidity section. According to the consensus statement, dry (RH < 20%) and humid air represent dramatically different stress conditions for PSCs[11,16], therefore 20% RH seems to be a reasonable threshold that also makes a natural reference point. The effect of humidity can then be expressed in the form of an acceleration factor, $A_{humidity}$ (RH) given by Eq. 3, where we based on previous results and simplicity have set the proportionality constant, $\gamma$, to 1. In practice, this model assumes that a device that is stable for 1 h at 80% RH would be stable for 4 h in dry air.

$$A_{humidity}(RH) = \begin{cases} \gamma \cdot \frac{RH}{20\%}, & \text{when } RH > 20\% \\ \gamma, & \text{when } RH \leq 20\% \end{cases} \quad (3)$$

The final environmental stress here considered is light intensity, which is known to affect PCE stability. One proposed mechanism for light-induced degradation is that superoxide generated by the transfer of photoexcited electrons to molecular oxygen reacts with the perovskite and results in rapid degradation[17,18]. In accelerated aging tests, the degradation rate is considered proportional to the light intensity[11,19]. It should, however, be noted that it is the photogenerated electron concentration, $n$, rather than the light intensity, $I_{in}$, that directly affects the degradation rate and there is not a simple linear relationship between the two variables. Timothy et al.[14] have proposed a power-law relation, $n \propto I_{in}^{0.7}$, which we here have adopted to determine the acceleration factor of light, $A_{light}(I)$. However, the light intensity is generally larger than 1 sun in an accelerated process and PSCs will work under different conditions at low light levels. We have therefore used 10 mW cm$^{-2}$ (0.1 sun) as a threshold to avoid the prediction of unrealistically long stabilities under dark conditions and 1 sun is used as the reference condition. Available stability data are almost always measured under either 1 sun or in the dark. The model here used assumes that 1 h of stability under 1 sun corresponds to 5 h of stability in the dark.

$$A_{light}(I) = \begin{cases} \frac{I^{0.7}}{(100mWcm^{-2})^{0.7}}, & \text{when } I > 10 mW\,cm^{-2} \\ 0.1^{0.7}, & \text{when } I \leq 10 mW\,cm^{-2} \end{cases} \quad (4)$$

Several available metrics try to capture device stability in one number. One approach is to state the PCE after a certain period, often 1000 h, $E_{1000h}$, or the time it takes to decay to a certain value, generally 80% of the initial efficiency, $T_{80}$. As $E_{1000h}$ requires a long test time and cannot interact with acceleration factors in accelerated aging tests, $T_{80}$ has become the most widely used indicator. For devices with better stability, the $T_{95}$ value is sometimes used instead.

One problem with the $T_{80}$ value is the burn-in phenomenon observed in many PSCs[20-22], where the efficiencies initially undergo a

quick increase or decrease, followed by a longer degradation region where the efficiencies drop more slowly (Fig. 1c). Reports are showing that the efficiency decay in the burn-in region often has some degree of reversibility and could be caused by an imbalanced ion distribution and charge accumulation[20,23]. From a stability point of view, it is thus the slower degradation after the burn-in phase that is of most interest as that capture the real degradation induced by environmental stress. With large efficiency changes in the burn-in region, there is also a problem that the $T_{80}$ value cannot give a good measure of the true degradation rate (Fig. 1c). To overcome those problems, the ISOS protocols suggest the use of the time to decay to 80% of the stabilized efficiency at the end of the burn-in region ($T_{S80}$) and an acceleration factor to describe the effect of the accelerated degradation process.

At the time of writing, the Perovskite Database contains stability data for 7419 devices. For most of the data, the total exposure time ($T_{end}$) and the efficiency at the end of the experiment ($E_{end}$) are available (7361 devices). There are also 1835 devices where a $T_{80}$ value is stated, but only in 95 cases are also the $T_{S80}$ values available (Supplementary Table 1, SI). To put all data on the same footing we have estimated the $T_{S80}$ for all devices. This is made by assuming that the degradation after the initial burn-in period follows a simple exponential decay and that the burn-in period is short compared to the total measurement time. This will not always be true, but it will be good enough for a first analysis. It enables an estimate of the $T_{S80}$ based on the reported efficiency at the end of the measurement and enables a direct comparison of all stability data. Details of the calculation method are provided in Supplementary Note 1 (SI).

Finally, we propose a single indicator, $T_{S80m}$, which takes into account the three major environmental stresses and predicts the $T_{S80}$ value under a reference condition. With 300 K, 20% RH and 1 sun illumination as the reference condition, devices tested under any temperature, humidity and light intensity can be described by $T_{S80m}$ as:

$$T_{S80m} = T_{S80} * A_{temperature} * A_{humidity} * A_{light} \quad (5)$$

## Hypothesis test

To explore what strategies and parameters that have resulted in improvements in device stability, we have here used the student's *t*-test[24], which gives a measure of the statistical significance of the difference found in the historic dataset.

The student's *t*-test assumes normally distributed data. According to the histogram (Fig. 2a) and the normal probability plot (Fig. 2b), the $T_{S80m}$ values are characterized by a log-normal distribution. Correspondingly, the log($T_{S80m}$) values (log for the natural logarithm) are normally distributed, proved by the histogram (Fig. 2c) and the normal probability plot (Fig. 2d), which enables the use of the standard statistical toolbox. However, this is not the case for the uncorrected $T_{S80}$ values. The distribution of those values (Supplementary Fig. 1, SI) is instead skewed towards longer stability times due to a large number of stability results under low environmental stress. Supplementary Table 2 (SI) gives the devices with the top 10 $T_{S80}$ values, which shows that all those devices were stored under dark and low temperatures, leading to a low degradation rate and very high $T_{S80}$ values. If no correction is made for the lack of stress factors, this will result in erroneous assessments of the stability-increasing strategies, so it is reasonable to use $T_{S80m}$ rather than $T_{S80}$ as the indicator.

Note that there are some very high $T_{S80m}$ values, stating more than 20 years of stability (Supplementary Fig. 2, SI). That may seem long, but is the result of stable devices measured under harsh conditions, as shown in Supplementary Table 3 (SI). The heuristics here used assumes that a 1 h of stability at 85 °C, 85% RH, and 1 sun illumination corresponds to 184 h of stability at 1 sun illumination and dry conditions at room temperature, and 1000 h at those conditions would thus

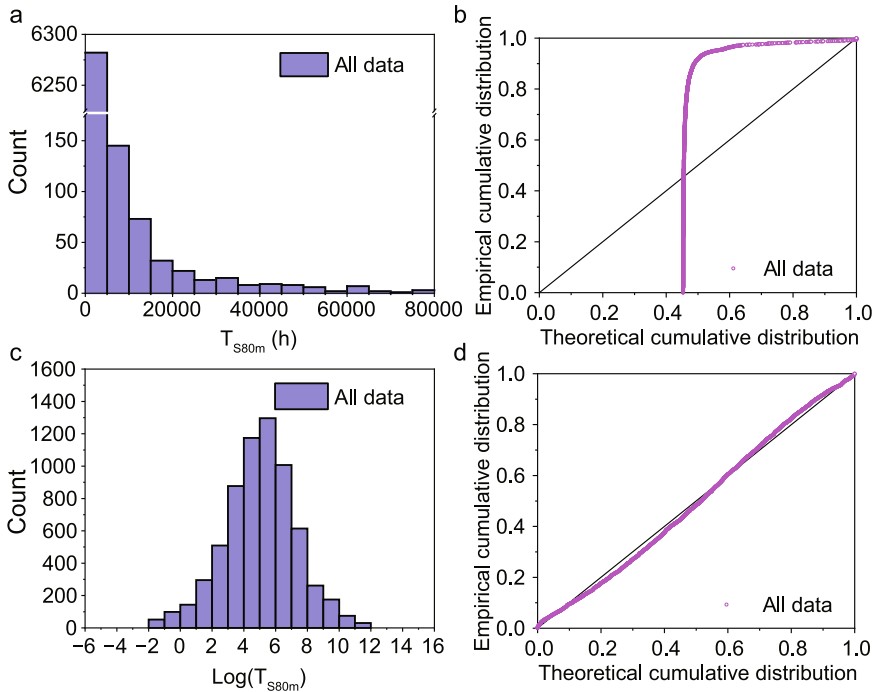

**Fig. 2 | An overview of the dataset. a** Histogram of $T_{S80m}$ values for the complete dataset up to 80,000 h. **b** Normal probability plot, i.e., the experimental cumulative distribution vs. the distribution for ideally normally distributed data. **c, d** The corresponding figures for log($T_{S80m}$) values (log for the natural logarithm).

correspond to over 20 years at our chosen standard conditions (i.e., 27 °C, 0% RH, and 1 sun illumination). The largest environmental stress impact behind this is the exponential dependence on temperature in the Arrhenius model (Eq. 2). Nevertheless, this is common to all data points and the large $T_{S80m}$ value is only a rough estimate, so they will not affect the conclusions.

After validating that the data distribution is normal, the data is divided into different samples, whereupon the student's $t$-hypothesis-test is applied to verify whether there is a statistically significant difference between the mean levels of the distributions. For distributions that are significantly different, we also get a maximum acceptable $T_A/T_B$ ratio which represents the multiple between the average $T_{S80m}$ values of the two distributions. For example, a $T_A/T_B$ ratio of 2 thus means that the average stability, expressed as a $T_{S80m}$ value, is two times larger for A than for B with 95% confidence. More details of the test process are described in Supplementary Note 2 (SI).

Below several modification strategies for different parts of PSCs, which can be distinguished in the Perovskite Database, are discussed and analyzed.

### Mixed-cation and mixed-anion perovskites

Perovskites have an $ABX_3$ structure, where a framework of corner-sharing $BX_6^{4-}$ octahedra surrounds larger A-site cations. A large range of perovskite compositions has been explored as light absorbers, but the most studied ones have $Cs^+$, $MA^+$, and/or $FA^+$ on the A-site, $Pb^{2+}$ and/or $Sn^{2+}$ on the B site, and $Cl^-$, $Br^-$, and/or $I^-$ on the X-site.

The by far most widely investigated perovskite is $MAPbI_3$[9,10]. $MAPbI_3$ is, however, often suffering from poor stability due to water-induced deprotonation and thermal-induced volatilization of $MA^+$[25]. Because of the simple composition, well-defined atomic positions in the crystal lattice and absence of cation alloying, pure $MAPbI_3$ PSCs are still significant to simplify mechanistic investigations[6,7,16]. $FAPbI_3$ is an alternative to $MAPbI_3$, which has a more suitable band gap for photovoltaic applications (~1.48 eV[26,27]) and better thermal stability[26]. Unfortunately, the $FA^+$ is slightly too large for the perovskite structure, which results in problems with structural instability and the formation

of a photoinactive yellow phase. $FAPbI_3$ can, however, be stabilized by mixing in other cat- and anions like $MA^+$, $Br^-$[28], and $Cs^+$[29], which results in mixed perovskite, where some of the most commonly used compositions have stoichiometries around $Cs_{0.05}FA_{0.79}MA_{0.16}PbBr_{0.51}I_{2.49}$ and $FA_{0.85}MA_{0.15}PbBr_{0.45}I_{2.55}$. All-inorganic cesium lead halide PSC is another branch of perovskite photovoltaics with potentially better thermal stability originating from the removal of the organic parts[30].

One way to categorize perovskites related to structural stability is to use the concept of the tolerance factor, $\alpha$, (Eq. 6) introduced by V.M. Goldschmidt in 1926, which is a way to evaluate ion radii mismatches[31].

$$\alpha = \frac{r_A + r_X}{\sqrt{2}(r_B + r_X)} \tag{6}$$

$r_A$, $r_B$, and $r_X$ are the radii of the A, B, and X ions in the perovskite $ABX_3$ structure. Since the tolerance factor varies among different ion compositions and proportions and gives close values for similar stoichiometries, it can be used as a feature to categorize perovskite compositions and analyze the relationship between compositions and stability.

Here, the total numbers of devices and the highest $T_{S80m}$ with respect to the perovskite tolerance factor and the publication date are visualized in the form of heat maps (Fig. 3a, b). Only three-dimensional (3D) perovskite compounds are included in this analysis. The Shannon radii of simple ions are used[32] and the effective radii of organic cations are according to Kieslich et al.[33] (listed in Supplementary Table 4, SI).

Figure 3a shows several horizontal lines with certain tolerance factor values, which represent specific compositions used over the years. The most commonly used perovskites and their calculated tolerance factors are listed in Supplementary Table 5 (SI). The $T_{S80m}$ heatmap (Fig. 3b) has a similar feature as Fig. 3a, where the widely used compositions mentioned above (FAPbI_3-based mix cation compositions, $MAPbI_3$ and all-inorganic compositions) are shown as continuous horizontal lines. There are several deep-color points on all these lines, indicating that all of these compositions are capable of excellent stability. The pure phases of $FAPbI_3$ and $CsPbI_3$ have some

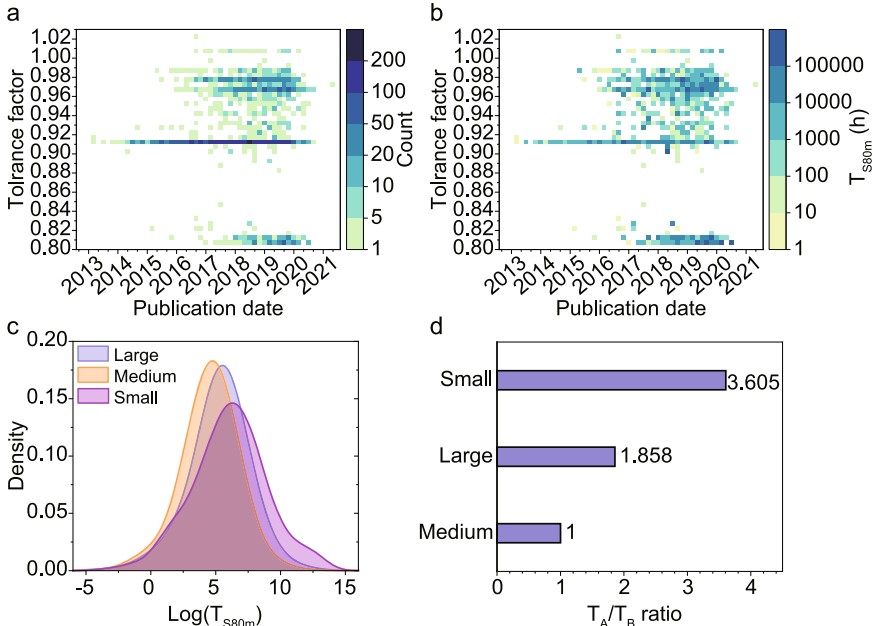

**Fig. 3 | The relationship between device stability and tolerance factors. a** A heatmap of the total numbers of devices with reported stability data with respect to the tolerance factor and the publication date. **b** A corresponding heatmap of the highest reported $T_{S80m}$ values. **c** The kernel density estimation of the log($T_{S80m}$) values for different tolerance factor regions (large, i.e., sample with tolerance factor $\alpha > 0.95$; medium, i.e., sample with $0.85 < \alpha < 0.95$; small, i.e., sample with $\alpha < 0.85$.) of three-dimensional perovskite devices without encapsulation. **d** A bar chart of the $T_A/T_B$ ratios (representing relative stability level) for the three different tolerance factor regions, where the ratio of the medium tolerance factors is set to 1.

data points demonstrating high stability, but have not reached the general stability level of the mixed ion compositions. FAPbI$_3$ and CsPbI$_3$ have the narrowest band gaps in lead-based hybrid and inorganic halide perovskites (without Sn$^{2+}$), which is beneficial for photovoltaic applications. The highest certified efficiencies of hybrid and all-inorganic PSCs are obtained with these two compositions, which are 25.7%[5] and 20.1%[34]. This demonstrates high intrinsic potential, but due to tolerance factor values near the edges of the feasible range both FAPbI$_3$ and CsPbI$_3$ suffer from structural instabilities and undesired phase transition under ambient conditions. Sn-based and Pb-Sn mixed perovskites, which are interesting due to their lower band gaps, show no impressive stability values in the heat maps.

However, it is difficult to draw accurate conclusions solely based on the most stable devices, therefore we have done a statistical analysis based on the complete dataset. The data were divided into three categories based on the tolerance factors, which are labeled: Large, i.e., sample with $\alpha > 0.95$, which mainly is FAPbI$_3$-based compositions; Medium, i.e., perovskites with a tolerance factor in the range $0.85 < \alpha < 0.95$, which to a large extent are MAPbI$_3$; Small, i.e., perovskites with a tolerance factor $\alpha < 0.85$, which to a large extent represent all-inorganic compositions.

The kernel density estimation of the distribution of log($T_{S80m}$) is given for the three categories in Fig. 3c. There is a large overlap between the three distributions. That is expected as the data is based on a decade's worth of publications of PCSs with various device architectures that have a large variation in terms of deposition procedures and choice of charge transport layers, etc. The interesting question is whether the historical dataset, with all its variability and the heuristics used to make the stability data comparable, can support that some ranges of perovskite compositions give intrinsically more stable PSC devices.

The first step toward answering that question is to check if all three log($T_{S80m}$) distributions follow a normal distribution. That turns out to be the case (Supplementary Figs. 3, 4, SI), which enables hypothesis testing based on the student's $t$-test. Taking the Medium sample as the reference sample, which has the smallest average, the

hypothesis test is done for each sample (Supplementary Table 6, SI), which gives a set of $T_A/T_B$ ratios depicted in Fig. 3d. The analysis shows that at the 95% confidence level there are statistically significant differences in stability based on the perovskites with different tolerance factors.

The analysis shows that the perovskites with the larger tolerance factors, i.e., FAPbI$_3$-based perovskites and related mixed compositions, are more stable than perovskites with medium tolerance factors, i.e., primarily MAPbI$_3$. It also shows that the average difference in time to failure as expressed in the $T_{S80m}$ value is a factor of 1.9. For the perovskites with smaller tolerance factors, i.e., primarily all-inorganic perovskites, the difference is even larger with 3.6 times longer average times to failure, which means that the all-inorganic perovskites are the most stable ones. Those perovskites also have a thicker tail at the high end of the stability distribution (Fig. 3c).

Those differences in stability could be verified in dedicated control experiments and are in line with what one may have suspected based on the published record values. An important result of this analysis is that it demonstrates that those intuitions hold even if all available data is considered, with all the possible variations found in the historic dataset, and that those intuitions are not the results of specific conditions in limited sets of experiments.

## Two-dimensional (2D) perovskites

The same analysis has been done to compare 2D, 3D, and 2D/3D mixed perovskites.

The 2D perovskites have attracted a lot of interest. In part because they greatly expand the structural and compositional range of possible perovskites, but also because of reports of improved environmental stability[35]. In 2D perovskites, single or multiple inorganic corner-sharing BX$_6^{4-}$ octahedra layers are sandwiched between large organic cation spacers. 2D perovskites generally have a formula of (A')$_m$A$_{n-1}$B$_n$X$_{3n+1}$, where A' is a large organic spacing cation, m is defined by the charge of A' and n equals the number of inorganic layers between the two neighboring spacers. The introduction of large organic cations improves the hydrophobicity of perovskite films,

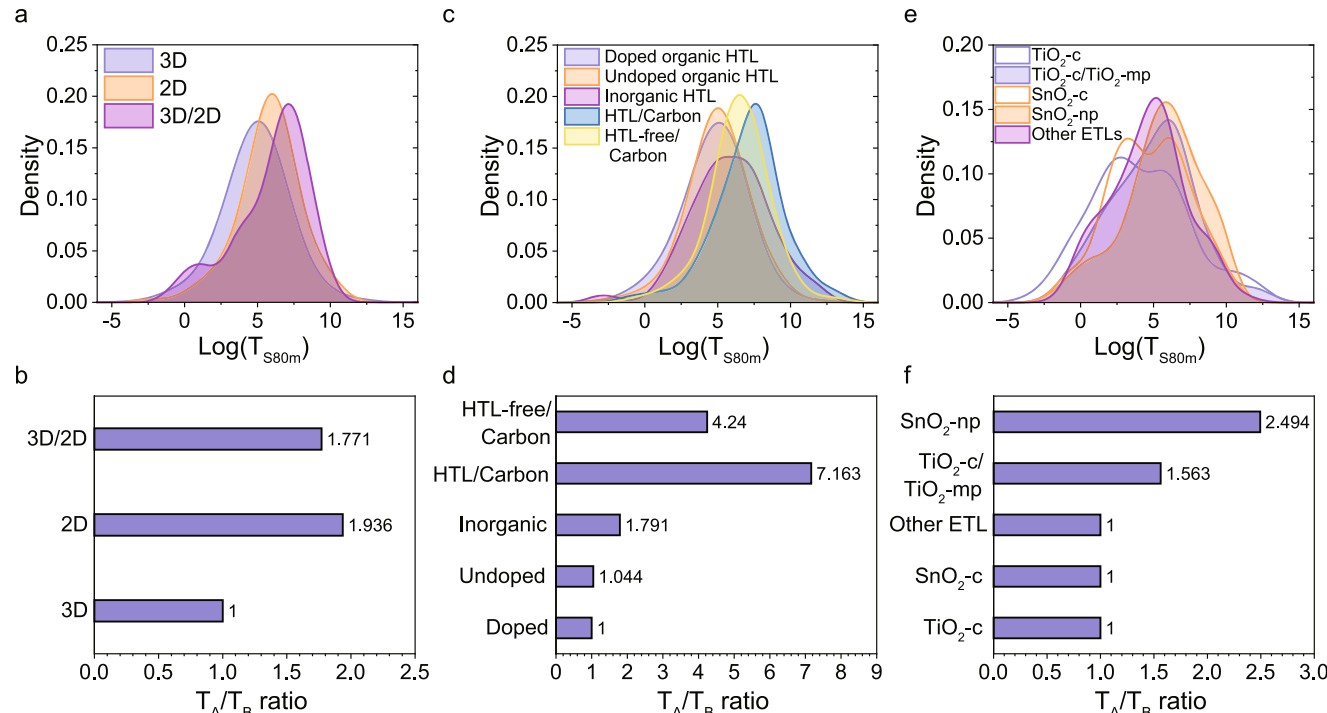

**Fig. 4 | The relationship between device stability and different functional layers.** The kernel density estimation of the $log(T_{S80m})$ values and the bar chart of $T_A/T_B$ ratios for unencapsulated devices with **a**, **b** different perovskite absorbers (3D, i.e., three-dimensional perovskites; 2D, i.e., two-dimensional perovskites; 3D/2D, i.e., three-dimensional perovskites with two-dimensional perovskite capping layers), where the ratio of three-dimensional devices is set to 1. **c**, **d** different hole transport layers (HTLs) and electrodes, where the ratio of doped organic HTL devices is set to 1. **e**, **f** different electron transport layers (ETLs), where the ratio of $TiO_2$ compact layer devices is set to 1. (np stands for nanoparticles, c stands for compact layers, and mp stands for mesoporous layers).

inhibits the volatilization of A-site organic cations and suppresses ion migration which enhances photostability[35,36]. On the other hand, they tend to have larger band gaps, stronger exciton binding energies, and hindered charge transport across the organic spacers, all of which may decrease device efficiencies[37]. Recently, 22.26% efficiency was obtained by optimizing the 2D perovskite absorber with $n = 5$, proving a success in the improvement of the efficiency of 2D perovskites[38]. Forming 3D perovskites with a 2D capping layer to combine the high efficiency of 3D perovskites and the stability of 2D perovskites is another approach that has gained recent attention[39–41].

The statistical analysis is in line with that 2D perovskites lead to more stable devices (Fig. 4a) and the hypothesis test indicates that the time to failure is about 1.9 longer than for the 3D perovskites (Fig. 4b and Supplementary Table 7, SI). The 3D/2D perovskites have a higher average, but they have a somewhat lower lifetime gain (a factor of 1.8 compared to the 3D perovskites). This contrast comes from the deviation from normal distribution caused by some devices in the range of low time to failure and the larger spread in reported values.

**Hole transport layers**

Besides the perovskite absorber layer, the choice of the hole transport layer (HTL) also influences the device stability. To give some more examples and investigate the device stability comprehensively, we also assess the stability improvement of different HTLs.

At present, PSCs with the highest efficiencies apply an n-i-p device structure and use doped 2,2′,7,7′-tetrakis[N,N-di(4-methoxyphenyl) amino]−9,9′-spirobifluorene (spiro-MeOTAD) as the HTLs[42–44] with lithium bis(trifluoromethanesulphonyl)imide (Li-TFSI) and 4-tert-butylpyridine (tBP) as dopants to increase the conductivity. However, HTLs based on Spiro-MeOTAD are suspected to have stability problems due to the degradation of Spiro-MeOTAD under light and heat and the hygroscopicity of dopants. Thus, poly[bis(4-phenyl)(2,4,6-trimethyl-phenyl)amine] (PTAA)[45], poly(3-hexylthiophene-2,5-diyl) (P3HT)[46], and

other new materials[47] are used to replace spiro-MeOTAD and dopants are removed to enhance stability. Inorganic HTLs including NiO[48], CuI[49], CuSCN[21], $Cu_2O$[50], and $CuCrO_2$[51] have also been widely studied due to their low fabrication cost, high conductivity and excellent thermal stability compared to organic HTLs.

No obvious difference has been seen for the organic HTLs. The kernel density estimation (Fig. 4c) shows that the two distributions of doped and undoped organic HTL almost completely overlap, with a $T_A/T_B$ ratio of less than 1.1 (Fig. 4d and Supplementary Table 8, SI), which means there is no obvious stability improvement by using dopant-free organic HTLs. For devices based on some of the most commonly used organic HTL, including spiro-MeOTAD, P3HT and PTAA, the analysis shows a 1.2 times stability gain for P3HT (Supplementary Fig. 8 and Supplementary Table 9, SI), and the kernel density estimation shows a peak of more stable devices with P3HT. That means that P3HT is a better choice among the organic HTLs. Inorganic HTLs, however, have a higher average and give 1.8 times longer device lifetime expressed as $T_{S80m}$.

Note that the devices discussed above are all based on metal electrodes and exclude carbon-based electrodes. That is because we found the carbon electrodes to have a significant effect on the stability improvement wherefore those were separated into their own category. The HTL-free devices with carbon electrodes show a significantly increased average with a $T_A/T_B$ ratio of 4.2 and when they are combined with HTLs, most of which are inorganic HTLs, the device stability expressed as $T_{S80m}$ is 7.2 times longer (Fig. 4d and Supplementary Table 9, SI). We think this is because inorganic HTLs not only protect perovskite absorber layers as carbon-based electrodes do, but also make up for the poor charge transport capacity of HTL-free devices which prevents charge accumulation.

In addition, devices with inorganic HTLs and/or carbon electrodes usually have lower efficiencies, so we also consider the balance between efficiency and stability. We use the product of efficiency gain

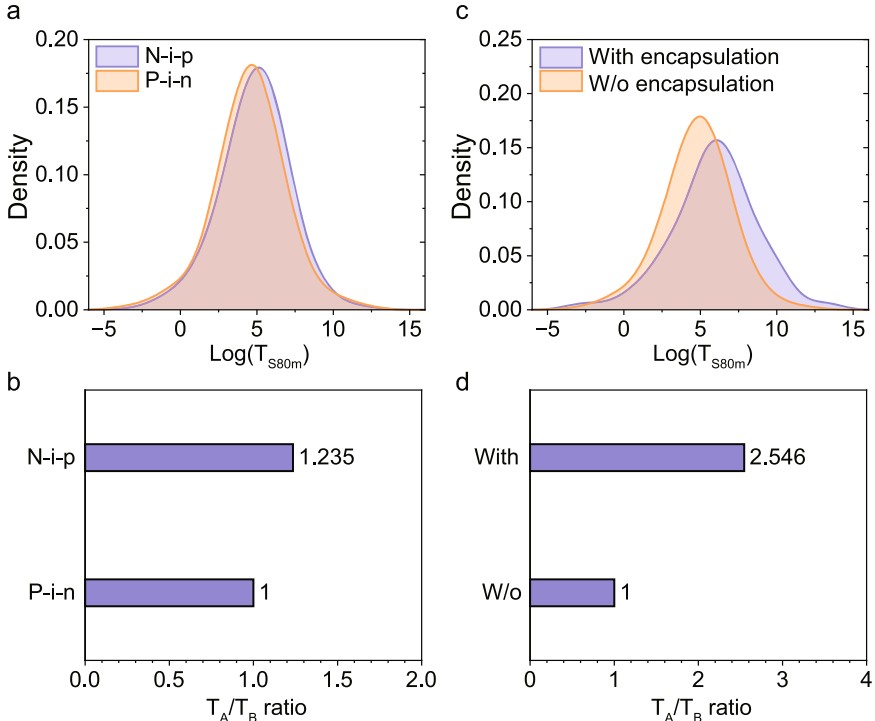

**Fig. 5 | The relationship between device stability and different device structures.** The kernel density estimation of the log($T_{S80m}$) values and the bar chart of $T_A/T_B$ ratios for **a**, **b** n-i-p and p-i-n structured devices, where the ratio of p-i-n devices is set to 1. **c**, **d** devices with and without encapsulation, where the ratio of unencapsulated devices is set to 1.

and stability gain as an indicator to compare the total energy outputs before the efficiency drops below 80%. The results are shown in Supplementary Fig. 11 and Supplementary Table 10 (SI), and we use the maximum of the smoothed probability density function as the efficiency values for calculation. The comparison shows that though inorganic HTLs improve device stability, the loss in efficiency (~4% drop in PCE) makes it less competitive, while the energy output capability of carbon-based devices is many times more than that of organic HTL devices. Together with the advantages of printability and simple preparation as previously reported[52], carbon electrodes can reduce the costs enormously.

### Electron transport layers

The most widely used electron transport layer (ETL) materials in n-i-p devices are $TiO_2$ and $SnO_2$, which account for 3799 and 912 of the 7419 devices, respectively. It has been reported that $TiO_2$-based n-i-p devices may undergo photo-induced degradation caused by oxygen vacancies and ultraviolet (UV) light-generated charge carriers[17,18,53,54]. Thus, device data under illumination without UV filters are collected, which decreases the dataset by 90%.

For the most common ETLs, the stability is affected by the layer morphology. The hypothesis test shows that $SnO_2$-np (np for nanoparticle) based devices have 2.5 times longer lifetime, $TiO_2$-c/$TiO_2$-mp (c for compact, mp for mesoporous) based devices have a factor of 1.6, while devices based on $TiO_2$-c, $SnO_2$-c and other ETLs have no obvious difference. Moreover, for those highest stability devices, $TiO_2$ is more likely to be chosen (Supplementary Table 3, SI), which means the fear that the photocatalytic activity of $TiO_2$ would be detrimental to perovskites under UV radiation may be less of a problem.

We also find that for the most $SnO_2$ based devices, the compact or nanoparticle ETLs are deposited with spin-coating, while there are several common deposition procedures for $TiO_2$. The statistical results (Supplementary Fig. 14 and Table 12, SI) show that devices with chemical bath deposited $TiO_2$-c layer and $TiO_2$-c/$TiO_2$-mp layers based on

spray-pyrolysis/ spin-coating have stability improvement with factors of 1.7 and 2.6 compared to spin-coated $TiO_2$-c based devices.

### Device configurations

Compared to the combinations of perovskite absorbers and transport layers, the analysis for device configurations is less complicated and the results are clearer.

The architectures of PSCs are generally divided into n-i-p and p-i-n structures. Due to the instability of doped spiro-MeOTAD, the stability issue of n-i-p structured PSCs is frequently mentioned. The statistical results show that n-i-p devices are slightly more stable than p-i-n devices (Fig. 5a) and the average $T_{S80m}$ value is 1.2 times longer (Fig. 5b and Supplementary Table 13, SI). That is contrary to expectations.

Considering that one of the sources of instability of n-i-p devices is spiro-MeOTAD, we limited the HTLs to spiro-MeOTAD for further analysis. The spiro-based n-i-p devices are still more stable than the p-i-n devices (Supplementary Fig. 16, SI), but the $T_A/T_B$ ratio is reduced to less than 1.1 (Supplementary Table 13, SI). We also focus on the most stable devices in each sample and give the lists of the top 10 n-i-p, p-i-n, and spiro-based n-i-p devices without encapsulation (Supplementary Tables 14–16, SI). The $T_{S80m}$ values of the top 10 unencapsulated n-i-p devices are slightly larger than p-i-n devices, but when the HTLs are limited to spiro-MeOTAD, the range decreases a lot, which means spiro-MeOTAD is not beneficial for very stable devices.

### Encapsulation

Encapsulation has proved to be a simple and effective strategy to improve the external stability of PSCs by preventing the penetration of moisture and oxygen[55–57] and to prevent lead leakage[58], which is a necessary part of commercialization.

Kernel density estimation (Fig. 5c) shows that encapsulated devices have higher average $T_{S80m}$ values and a thicker tail attributed to a few exceptionally stable devices. The hypothesis test also shows that

that the average lifetime of encapsulated devices is 2.5 times longer (Fig. 5d and Supplementary Table 16, SI).

Since encapsulation and carbon electrodes can both protect the devices to give a significant improvement in device stability, we also investigate their differences and synergies. The statistical analysis (Supplementary Fig. 19, SI) shows that carbon electrodes have a stronger effect on stability improvement than encapsulation, with a $T_A/T_B$ ratio of 1.6 (Supplementary Table 18, SI). For encapsulated carbon-based devices, though the kernel density estimation seems to show better stability, the hypothesis test gives no difference for the combination of carbon electrodes and encapsulation over a single strategy. That comes from the deviation from normal distribution caused by some devices in the range of low $T_{S80m}$ and the small sample size.

### Discussion on uncertainty and reproducibility

The indicator $T_{S80m}$ is calculated by converting three main environmental stresses, temperature, humidity and light intensity to separate acceleration factors and multiplying them with $T_{S80}$. Uncertainty will come from the co-dependencies between different stressors, the range of parameters ($E_a$ in $A_{temperature}$ and $\gamma$ in $A_{humidity}$) and the chosen reference condition.

For the range of parameters ($E_a$ and $\gamma$), the different values will make $T_{S80m}$ more sensitive or less to the environmental stresses. For example, with a larger $E_a$ value, one device will achieve a higher $T_{S80m}$ from $A_{temperature}$. Supplementary Tables 19, 20 show that the average of $T_{S80m}$ is positively related to both $E_a$ and $\gamma$. However, only $E_a$ influences the hypothesis test results because of the exponential relationship, while the change of $\gamma$ has the same effect on all the devices, which keeps the results the same. Thus, reasonable parameter values are needed for the lifetime estimation, but the hypothesis test is less affected.

In addition, different reference conditions will not affect the conclusions. In accelerated degradation tests, hundreds of hours of tests under harsh conditions are used to predict tens of years of the device lifetime. $T_{S80m}$ predicts the lifetime under the reference conditions (27 °C, 0% RH, and 1 sun illumination), so the value of $T_{S80m}$ is usually much larger than common testing results. We also choose 85 °C, 85% RH and 1 sun illumination as the reference conditions and recalculate $T_{S80m}$. The results show that all the data points only shift to smaller values without change in shape (Supplementary Fig. 21, SI), and the hypothesis test conclusion about the tolerance factor remains the same (Supplementary Table 21).

The detail of the hypothesis test method is described in Supplementary Note 2 (SI). An accepted hypothesis, which means there is a statistically significant difference between two samples, requires large sample sizes, small variances and large average differences. Thus, the limitation of data (small data sizes and large variances) tends to give an unaccepted hypothesis. Nevertheless, the strategies which show obvious stability improvement are still credible.

As mentioned above, the Perovskite Database contains stability data for 7419 devices with publication data from 2012.08.21 to 2021.05.21 at the time of writing. Note that only a small number of publications are included, but the dataset is sufficient to draw conclusions that are consistent with the current state of the field. However, the research focus of the PSC field changes over time (e.g., the change of mainstream perovskite compositions), so the conclusions are not always true and may be overturned in the future. Time-dependent statistical analysis is needed to draw dynamic conclusions, which is beyond the scope of this work.

## Discussion

In summary, we proposed a feasible unified indicator, $T_{S80m}$, to assess the stability of devices under different test conditions, which takes into account temperature, humidity, and light intensity, thus enabling direct comparison of stabilities measured under different conditions.

With the indicator and a historical dataset of over 7000 stability measurements found in the Perovskite Database, we have performed the statistical analysis and assessed the significance of different strategies for improving device stability based on hypothesis testing.

The results indicate that n-i-p devices are more stable than p-i-n devices. All-inorganic perovskite compositions are more stable than $FAPbI_3$-based mixed ion compositions, which in turn are more stable than $MAPbI_3$-based devices. Reducing the perovskite dimensionality, either by using 2D perovskites or 3D/2D stacked structures can improve the stability further. In terms of stability, inorganic HTLs seem to be superior to their organic counterparts, regardless of whether the latter are doped or not. Switching to carbon electrodes does, however, outcompete everything else, especially if combined with an inorganic buffer layer. Unfortunately, top efficiencies lag behind those solutions. For ETLs, $SnO_2$ appears to have a slight stability advantage over $TiO_2$ and other ETLs, but the difference is sufficiently small to assume that photocatalytic degradation of the perovskite at the $TiO_2$ surface is not a serious problem. Encapsulation does not surprisingly have a significant positive effect on stability, but switching to carbon electrodes seems to have a far larger positive impact.

Combining those findings indicates that, if one searches for stable devices, a good place to start would be to look for n-i-p devices with a device structure of substrate/$SnO_2$/all-inorganic perovskite absorber/ 2D capping layer/inorganic HTL/carbon electrode. Carbon electrodes can effectively block moisture and oxygen, and moisture-sensitive and thermally stable all-inorganic perovskites could be well protected. Encapsulation can further improve stability. Considering that devices with inorganic HTLs and carbon electrodes so far have lower efficiencies, a structure of substrate/$SnO_2$/all-inorganic perovskite absorber/2D capping layer/P3HT(PTAA)/metal electrode with encapsulation would for now be a good suggestion for a device combining stability and efficiency. The 2D capping layer can be replaced by other stable materials and spiro-MeOTAD is not recommended. Interestingly, there is still no device containing all those options reported in the Perovskite Database, which makes an obvious suggestion for further experimental studies.

This work provides a feasible example of the application of the statistical method for PSC stability assessment based on a large open database of historical data and provides a reference for further data mining projects. If we look forward, there is much that can be improved.

Most important is the data quality and development of more accurate models for describing the device stability under environmental stress. The degradation rates of PSCs are influenced by temperature, humidity, light intensity, circuit bias, and environmental stress cycles, and PSCs have several degradation ways including perovskite and transport layer decomposition, perovskite phase transition, as well as ion migration and penetration. The complex degradation process and multi-factor synergistic effect make it difficult to precisely predict device performance with simple models. For higher accuracy predictions, the degradation mechanisms thus need to be clarified and device degradation behaviors need to be quantitatively reported rather than only qualitatively described. With accurate models, corresponding stability indicators can be set up for more reliable statistical analysis and the establishment of more refined stability test standards in the perovskite field.

Secondly, more data is needed to draw more solid and detailed conclusions in the future. In addition, due to the various kinds of materials used as HTLs, ETLs and interfacial modifiers, it would be valuable if those could be labeled according to their energy levels, conductivity, molecule structural pattern, active groups, and their action mechanism in addition to their names. Similar operations are also required when recording the HTLs, ETLs, and perovskite absorber additives. With proper labels, statistical analysis will be easier to perform and inferred conclusions more general and robust.

## Methods

Data are downloaded from the Perovskite Database Project on 2022.01.18. From this data all the parameters of devices with stability measured are extracted for statistical analysis. The file was named datam.csv and there are 7419 devices in total. Student's $t$-hypothesis-test method is used for the comparison between different strategies, and the confidence level is set to 95%. The details of the method are described in Supplementary Note 2.

All the code for generating the analysis and the plots is written with Wolfram Mathematica 13.

## Data availability

The computed tolerance factors, $T_{S80}$, $A_{temperature}$, $A_{humidity}$, $A_{light}$, and $T_{S80m}$ values generated in this study together with the extracted origin stability data have been deposited in a public GitHub repository, which is available at https://github.com/NK-ZZhang/PSC-stability.git. It is also available in the Zenodo repository (https://doi.org/10.5281/zenodo.7345315)[59].

## Code availability

All the codes for generating the analysis and the plots are available in the GitHub repository https://github.com/NK-ZZhang/PSC-stability.git as well as the Zenodo repository (https://doi.org/10.5281/zenodo.7345315)[59].

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

## Acknowledgements

J.L. acknowledges the funding support from the National Key R&D Program of China (2021YFF0501900 and 2019YFE0123400), the Excellent Young Scholar Fund from the National Science Foundation of China (22122903), the Tianjin Distinguished Young Scholar Fund (20JCJQJC00260), and the Haihe Laboratory of Sustainable Chemical Transformations.

## Author contributions

J.L. supervised the project. Z.Z. designed the data analysis method and executed the code writing. J.L. provided the computing platform. J.L., T.J., Z.Z., and H.W. contributed to the discussion of the results and the writing of the manuscript.

## Competing interests

The authors declare no competing interests.
