## [Peer Review File · Nature Communications]

Big Data Driven Perovskite Solar Cell Stability AnalysisReviewer #1 (Remarks to the Author):

In this paper, the authors proposed a single indicator TS80m for stability evaluation of PSCs under various testing conditions and analyzed the collected stability data in the Perovskite Database Project to assess the influence of different perovskite compositions and device configurations on PSC stability. It is interesting to draw conclusions with all available historic data rather than experimental results with specific conditions, and the analysis method can attract many researchers in the same field. It would be a valuable contribution to the field. I recommend it for publication after minor revisions. Please find below the specific comments and suggestions.

1. In section 2.2 paragraph 2, the authors said that 1 h of stability at 85°C, 85 % RH, and 1 sun illumination corresponds to 184 h of stability in dark and dry conditions at room temperature. According to the formulas and tables given by the authors, there may be some miscalculations. The authors should check them.

2. In section 4.1 paragraph 2, the authors compared the stability of devices with different HTLs and electrodes. Since inorganic HTLs and carbon electrodes were proved to affect the efficiency of PSCs, a comparison of efficiency and some discussions should be added to verify whether the stability gain balances the stability loss.

3. In section 4.1 paragraph 2, the authors compared the stability of devices with spiro, P3HT, and PTAA. How did the authors deal with multi-HTL devices? If these devices were misclassified, the results would be biased.

4. In section 4.2 paragraph 2, the authors compared the stability of devices with different ETLs, mainly SnO₂ and TiO₂. Though there are statistical differences in the results, solid conclusions are hard to reach because of the small t_a/t_b ratio and deformed distribution curves. Since the same ETL material with different structures and deposition methods can lead to different stability, the authors could give a more detailed discussion accordingly.

5. Though the authors used all available historic data to perform statistical analysis, some groups still have small data sizes and large variances. How do these affect the statistical results and the reliability of the conclusions? The authors should add a discussion in the main text.

Reviewer #2 (Remarks to the Author):

This work leverages a large, open-access perovskite dataset that has recently been compiled from literature, and attempts to draw statistical conclusions about factors influencing device stability.

The main contribution of the paper is the introduction of a new stability metric (Ts80m), which accounts for temperature, humidity and light intensity, and therefore allows measurements made under different conditions to be compared. Using this metric to analyse the dataset, the authors draw several conclusions, the most noteworthy of which is, perhaps, the (fully expected) dependence of stability on the tolerance factor.

As the authors themselves admit, perovskite degradation is a complex process where different stressors interact with each other. For instance, in Sn-based perovskites, it is known that the presence of humidity and oxygen sets up a vicious cycle of oxidation that kills the luminescence efficiency. Similar co-dependencies have been reported for most other perovskites. For this reason, a simplistic model that treats stressors as separate, and ignores oxygen and biasing conditions, is not satisfactory. For this metric to be convincing, either:

1. Gather experimental data to show that the effects of different stressors don't influence one other in perovskite stability testing.
2. (Easier) Look into your dataset to see if you have cells for which humidity, illumination and temperature have been reported, and use them to demonstrate the validity of equation 5.

In my opinion, the above issue is a major one and prevents the model from being used as more than a crude metric. Even as a crude metric, it is not very convincing. In fig 2c, majority of the cells are said to have a Ts80m of many hundred hours. In practice, very few cells have such performance today. Thus Ts80m largely overestimates stability, probably because it fails to consider how stressors reinforce each other.

The range of values that gamma and activation energy can take is likely to introduce some uncertainty in the calculations. The authors should report what this uncertainty is.

Furthermore, there is nothing new that the analysis reveals. Almost everything is well-known: all-inorganics are more stable than hybrid perovskites, n-i-p is more stable than p-i-n historically (now being overturned), 2D/3D heterostructures more stable than 3D...

The idea that the tolerance factor strongly determines stability is perhaps shown here for the first time.

The authors have provided all necessary details for the paper to be reproduced.

The stability model developed by the authors is too simplistic to account for the complex nature of perovskite degradation where stressors reinforce each other. Furthermore, the analysis validates what is already known in the field, and does not offer anything new. Since these are major issues with the paper, perhaps difficult to correct without major re-writing, I suggest that it be submitted to a different journal.

Reviewer #3 (Remarks to the Author):

This manuscript proposes big data driven perovskite solar cell stability analysis. The topic is interesting, and certainly consistent with the contents to be proposed to the readers of "Nature Communications". Moreover, the manuscript is well written and can be read with pleasure: this represents an important aspect in the current scenario of publications in international journals. Overall, I think that this manuscript has to be accepted, but the Authors should take into account the following minor revisions (in terms of bibliographic updates, grammar corrections and content deepening):

- It is difficult to review a manuscript without page numbers!!!
- Detailed revisions: I spent several hours reading this manuscript, and Authors are asked to follow carefully the attached PDF file where I highlighted some points to be addressed. The attached file also contains language mistakes and typos; some questions related to manuscript contents could also be present and Authors must consider them properly before submitting the revised manuscript. A point-by-point reply is required when the revised files are submitted.
- The Introduction should give a wider overview on the present scenario related to hybrid photovoltaics, both in terms of recently published reviews and research articles. In particular, emerging sustainable, integrated and unconventional PV devices are missing and a paragraph on this topic is highly suggested to be added in the Introduction. Authors are invited to go through the literature published in the last six months on these issues, and also on concepts developed some years ago in this field. Some of them are also mentioned in the above mentioned PDF file.
- Authors should provide a clear explanation on the experimental error of the proposed research work. In particular, reproducibility of the phenomena described in the manuscript should be clearly stated in the "Results and Discussion" section; besides, some notes in the "Materials and Methods" section should be added highlighting which kind of experimental approach has been followed to check the reproducibility of the proposed system, the latter being of noteworthy importance in the present research field.

In their review of the first version of this manuscript, reviewer #3 added some comments to the manuscript file. These comments were forwarded to the authors, who replied as included in this Peer Review File.

Point-by-point reply to reviewer comments

We thank all the reviewers for their valuable comments, which we have used to improve our work. Please find below the point-by-point reply to the comments, with the reply in blue color. The revisions made in the revised manuscript are highlighted in yellow color.

Reviewer 1:

In this paper, the authors proposed a single indicator TS80m for stability evaluation of PSCs under various testing conditions and analyzed the collected stability data in the Perovskite Database Project to assess the influence of different perovskite compositions and device configurations on PSC stability. It is interesting to draw conclusions with all available historic data rather than experimental results with specific conditions, and the analysis method can attract many researchers in the same field. It would be a valuable contribution to the field. I recommend it for publication after minor revisions. Please find below the specific comments and suggestions.

We thank the reviewer for the positive comments.

Question 1:

In section 2.2 paragraph 2, the authors said that 1 h of stability at 85°C, 85% RH, and 1 sun illumination corresponds to 184 h of stability in dark and dry conditions at room temperature. According to the formulas and tables given by the authors, there may be some miscalculations. The authors should check them.

We thank the reviewer for checking our work in detail and pointing out this.

The value is calculated by the equation,

$$A = A_{\text{temperature}} * A_{\text{humidity}} * A_{\text{light}} = e^{\frac{-E_a}{k_B} \left(\frac{1}{T} - \frac{1}{300} \right)} \cdot \gamma \cdot \frac{RH}{20\%} \cdot \frac{I^{0.7}}{(100mW \cdot cm^{-2})^{0.7}}$$

The reference conditions in the equation are 27 °C, 0% RH and 1 sun illumination, and the value of A is 184, which is correct. We miswrote 1 sun illumination as dark, which we have fixed now.

Revised manuscript:

“The heuristics here used assumes that a 1 h of stability at 85 °C, 85% RH, and 1 sun illumination corresponds to 184 h of stability at 1 sun illumination and dry conditions at room temperature, and 1000 hours at those conditions would thus correspond to over 20 years at our chosen standard conditions (i.e., 27 °C, 0% RH and 1 sun illumination).”

Question 2:

In section 4.1 paragraph 2, the authors compared the stability of devices with different HTLs and electrodes. Since inorganic HTLs and carbon electrodes were proved to affect the efficiency of PSCs, a comparison of efficiency and some discussions should be added to verify whether the stability gain balances the stability loss.

Following the reviewer's suggestion, we have compared the efficiencies with different HTLs, where the inorganic HTL devices have a ~4% drop compared to the organic HTL devices, and the carbon-based devices have a ~7% drop overall. We use the product of efficiency and stability gain to compare the total energy output of different devices before the efficiency drops to 80%. For inorganic HTL devices, the loss in efficiency makes them less competitive. However, carbon-based devices still have 2-4 times more energy outputs than organic HTL devices despite the reduced efficiency to a half, and that makes carbon electrode a promising candidate for commercialization. A detailed discussion has been added in the manuscript.

Revised manuscript:

“In addition, devices with inorganic HTLs and/or carbon electrodes usually have lower efficiencies, so we also consider the balance between efficiency and stability. We use the product of efficiency gain and stability gain as an indicator to compare the total energy outputs before the efficiency drops below 80%. The results are shown in Supplementary Fig. 11 and Supplementary Table 10 (SI), and we use the maximum of the smoothed probability density function as the efficiency values for calculation. The comparison shows that though inorganic HTLs improve device stability, the loss in efficiency (~4% drop in PCE) makes it less competitive, while the energy output capability of carbon-based devices is many times more than that of organic HTL devices. Together with the advantages of printability and simple preparation as previously reported⁶², carbon electrodes can reduce the costs enormously.”

Supplementary Fig. 11 The histograms and kernel density estimation of efficiency values for devices with different HTLs without encapsulation. **a**, doped organic HTL. **b**, undoped organic HTL. **c**, inorganic HTL. **d**, HTL/Carbon. **e**, HTL-free/Carbon. **f**, kernel density estimation.

Supplementary Table 10. The balance between efficiency and stability of doped organic HTL, undoped organic HTL, inorganic HTL, and HTL-free devices.

Sample	Efficiency (%)	Efficiency gain	Stability gain (T_A/T_B ratio)	Output gain
Doped organic HTL	17.90	1	1	1
Undoped organic HTL	17.10	0.955	1.044	0.997
Inorganic HTL	13.47	0.753	1.791	1.349
HTL/Carbon	9.944	0.556	7.163	3.983
HTL-free/Carbon	10.50	0.587	4.240	2.489

Question 3:

In section 4.1 paragraph 2, the authors compared the stability of devices with spiro, P3HT, and PTAA. How did the authors deal with multi-HTL devices? If these devices were misclassified, the results would be biased.

We thank the reviewer for the question.

The multi-HTL devices were included in the statistical samples according to the HTLs they have, and the combined effects of multi-HTLs reduced the difference between samples. In the revised manuscript, we focus on the single-HTL devices, and this reflects the role of HTL materials better. P3HT devices are still the most stable, which is the same as the previous conclusion, and the hypothesis test shows more obvious difference with a T_A/T_B ratio of ~ 1.2 . A detailed discussion has been added in the revised manuscript.

Revised manuscript:

“For devices based on some of the most commonly used organic HTL, including spiro-MeOTAD, P3HT, and PTAA, the analysis shows a 1.2 times stability gain for P3HT (Supplementary Fig. 9 and Supplementary Table 9, SI), and the kernel density estimation shows a peak of more stable devices with P3HT. That means that P3HT is a better choice among the organic HTLs.”

Supplementary Fig. 9 The histograms and kernel density estimation of $\log(T_{80m})$ values for devices with different HTLs without encapsulation. **a**, Spiro-MeOTAD. **b**, P3HT. **c**, PTAA. **d**, kernel density estimation.

Supplementary Fig. 10 The normal probability plots of $\log(T_{S80m})$ values for devices with different HTLs without encapsulation. **a**, Spiro-MeOTAD. **b**, P3HT. **c**, PTAA.

Supplementary Table 9. Statistical results of Spiro-MeOTAD, PTAA, and P3HT devices.

Group	Sample	Average	Variance	Size	H_0 accepted	μ	T_A/T_B ratio
1	PTAA	4.967	7.388	161	No	-	-
	Spiro-Me OTAD	4.731	4.690	2930			
2	P3HT	5.267	10.30	128	Yes	0.206	1.228
	Spiro-Me OTAD	4.731	4.690	2930			

Question 4:

In section 4.2 paragraph 2, the authors compared the stability of devices with different ETLs, mainly SnO₂ and TiO₂. Though there are statistical differences in the results, solid conclusions are hard to reach because of the small t_a/t_b ratio and deformed distribution curves. Since the same ETL material with different structures and deposition methods can lead to different stability, the authors could give a more detailed discussion accordingly.

We thank the reviewer for the suggestion.

We have regrouped the data into TiO₂-c, TiO₂-c/TiO₂-mp, SnO₂-c, SnO₂-np and other ETLs (c for compact, mp for mesoporous, np for nanoparticle). The results show that devices based on TiO₂-c, SnO₂-c and other ETLs have no difference, while SnO₂-np devices have better stability than TiO₂-c/TiO₂-mp. However, for those highest stability devices, TiO₂ is more likely to be chosen. For TiO₂ with different deposition procedures, chemical bath deposited TiO₂-c layer and TiO₂-c/TiO₂-mp layers based on spray-pyrolysis/ spin-coating have obvious stability improvement than spin-coated TiO₂-c layers. A detailed discussion has also been added in the manuscript.

Revised manuscript:

“For the most common ETLs, the stability is affected by the layer morphology. The hypothesis test shows that SnO₂-np (np for nanoparticle) based devices have 2.5 times longer lifetime, TiO₂-c/TiO₂-mp (c for compact, mp for mesoporous) based devices have a factor of 1.6, while devices based on TiO₂-c, SnO₂-c, and other ETLs have no obvious difference. Moreover, for those highest stability devices, TiO₂ is more likely to be chosen (Supplementary Table 2, SI), which means the fear that the photocatalytic activity of TiO₂ would be detrimental to perovskites under UV radiation may be less of a problem.

We also find that for the most SnO₂ based devices, the compact or nanoparticle ETLs are deposited with spin-coating, while there are several common deposition procedures for TiO₂. The statistical results (Supplementary Fig. 14 and Supplementary Table 12) show that devices with chemical bath deposited TiO₂-c layer and TiO₂-c/TiO₂-mp layers based on spray-pyrolysis/spin-coating have stability improvement with factors of 1.7 and 2.6 compared to spin-coated TiO₂-c based devices.”

Figure 4. The kernel density estimation of the $\log(T_{80m})$ values and the bar chart of T_A/T_B ratios for unencapsulated devices with **a-b**, different perovskite absorbers, where the ratio of 3D devices is set to 1. **c-d**, different HTLs and electrodes, where the ratio of doped organic HTL devices is set to 1. **e-f**, different ETLs, where the ratio of TiO₂-c devices is set to 1.

Supplementary Fig. 12 The histograms of $\log(T_{S80m})$ values for devices with different ETLs without encapsulation. **a**, $\text{TiO}_2\text{-c}$. **b**, $\text{TiO}_2\text{-c}/\text{TiO}_2\text{-mp}$. **c**, $\text{SnO}_2\text{-c}$. **d**, $\text{SnO}_2\text{-np}$. **e**, Other ETLs.

Supplementary Fig. 13 The normal probability plots of $\log(T_{S80m})$ values for devices with different ETLs without encapsulation. **a**, $\text{TiO}_2\text{-c}$. **b**, $\text{TiO}_2\text{-c}/\text{TiO}_2\text{-mp}$. **c**, $\text{SnO}_2\text{-c}$. **d**, $\text{SnO}_2\text{-np}$. **e**, Other ETLs.

Supplementary Fig. 14 The histograms and kernel density estimation of efficiency values for devices with different HTLs without encapsulation. **a**, TiO₂-c by spin-coating. **b**, TiO₂-c by spray-pyrolysis. **c**, TiO₂-c by CBD. **d**, TiO₂-c/ TiO₂-mp by spin-coating/ spin-coating. **e**, TiO₂-c/ TiO₂-mp by spray-pyrolysis/ spin-coating. **f**, kernel density estimation.

Supplementary Fig. 15 The normal probability plots of $\log(T_{S80m})$ values for devices with different ETLs without encapsulation. **a**, TiO₂-c by spin-coating. **b**, TiO₂-c by spray-pyrolysis. **c**, TiO₂-c by CBD. **d**, TiO₂-c/ TiO₂-mp by spin-coating/ spin-coating. **e**, TiO₂-c/ TiO₂-mp by spray-pyrolysis/ spin-coating.

Supplementary Table 11. Statistical results of SnO₂, TiO₂, and other ETL devices.

Group	Sample	Average	Variance	Size	H ₀ accepted	μ	T _A /T _B ratio
1	TiO ₂ -c /TiO ₂ -mp	4.957	8.187	413	Yes	0.447	1.563
	TiO ₂ -c	4.089	11.17	212			
2	SnO ₂ -c	4.722	7.131	78	No	-	-
	TiO ₂ -c	4.089	11.17	212			
3	SnO ₂ -np	5.619	7.021	105	Yes	0.914	2.494
	TiO ₂ -c	4.089	11.17	212			
4	Other ETLs	4.649	6.650	87	No	-	-
	TiO ₂ -c	4.089	11.17	212			

Supplementary Table 12. Statistical results of TiO₂-c (spin), TiO₂-c (spray), TiO₂-c (CBD), TiO₂-c/ TiO₂-mp (spin/spin), TiO₂-c/ TiO₂-mp (spray/spin).

Group	Sample	Average	Variance	Size	H ₀ accepted	μ	T _A /T _B ratio
1	TiO ₂ -c (spray)	4.200	24.39	29	No	-	-
	TiO ₂ -c (spin)	3.903	9.424	127			
2	TiO ₂ -c (CBD)	5.425	10.34	35	Yes	0.542	1.719
	TiO ₂ -c (spin)	3.903	9.424	127			
3	TiO ₂ -c/ TiO ₂ -mp (spin/spin)	4.447	7.829	177	No	-	-
	TiO ₂ -c (spin)	3.903	9.424	127			
4	TiO ₂ -c/ TiO ₂ -mp (spray/spin)	5.390	7.998	233	Yes	0.956	2.601
	TiO ₂ -c (spin)	3.903	9.424	127			

Question 5:

Though the authors used all available historic data to perform statistical analysis, some groups still have small data sizes and large variances. How do these affect the statistical results and the reliability of the conclusions? The authors should add a discussion in the main text.

We thank the reviewer for the suggestion.

According to the hypothesis test method, small data sizes and large variances will lead to unaccepted hypotheses. Nevertheless, the strategies which show obvious stability improvement are still credible. We have added a discussion in the manuscript.

Revised manuscript:

“The detail of the hypothesis test method is described in Supplementary note 2 (SI). An accepted hypothesis, which means there is a statistical significant difference between two samples, requires large sample sizes, small variances and large average difference. Thus, limitation of data (small data sizes and large variances) tends to give an unaccepted hypothesis. Nevertheless, the strategies which show obvious stability improvement are still credible.”

Reviewer 2:

This work leverages a large, open-access perovskite dataset that has recently been compiled from literature, and attempts to draw statistical conclusions about factors influencing device stability.

The main contribution of the paper is the introduction of a new stability metric (T_{S80m}), which accounts for temperature, humidity and light intensity, and therefore allows measurements made under different conditions to be compared. Using this metric to analyse the dataset, the authors draw several conclusions, the most noteworthy of which is, perhaps, the (fully expected) dependence of stability on the tolerance factor.

We thank the reviewer for the comments, which help improve the quality of our manuscript.

Question 1:

As the authors themselves admit, perovskite degradation is a complex process where different stressors interact with each other. For instance, in Sn-based perovskites, it is known that the presence of humidity and oxygen sets up a vicious cycle of oxidation that kills the luminescence efficiency. Similar co-dependencies have been reported for most other perovskites. For this reason, a simplistic model that treats stressors as separate, and ignores oxygen and biasing conditions, is not satisfactory. For this metric to be convincing, either:

- (1) Gather experimental data to show that the effects of different stressors don't influence one other in perovskite stability testing.
- (2) (Easier) Look into your dataset to see if you have cells for which humidity, illumination and temperature have been reported, and use them to demonstrate the validity of equation 5.

Though T_{S80m} is a single indicator for rough estimation, it enables a simple and effective assessment of PSC device stability comparison and succeeds to lead to correct and specific conclusions. To further refine the indicator, a more accurate and general mathematical model that contains all the testing parameters and degradation processes is needed. However, most of publications focusing on the degradation processes only investigate the degradation pathways and products. For the works with quantitative results of degradation rates, they only focus on specific compositions and limited testing conditions. Overall, there is still no unified model that cover all testing conditions and device types so far, and that is exactly what we are going to investigate next. But for now, T_{S80m} is the most effective indicator we can get based on existing works in the field. Moreover, there are lots of work to do with the PSC stability standard and statistical analysis and we aim to provide a feasible example to induce more contributions in this topic.

We have also tried to look into the dataset to find if there are some groups of data that can validate the co-dependencies. To do this, the data in one group need to meet some requirements.

(1) The data come from the same publication. Due to the variance between different laboratories, material suppliers, experimental conditions, instruments, and some other hidden variables, devices usually have different performances even if all reported parameters (compositions, device structures, preparation processes, testing conditions, etc.) are the same. If we choose data from the same publication, the differences can be eliminated as much as possible.

(2) The same composition, device structure and preparation process are used. As widely reported, the device performance is influenced by these parameters.

(3) The data contain specific combinations of testing conditions. For example, if we consider the co-dependencies between temperature (T) and humidity (H), a valid group of data should contain results under at least four environmental conditions, (T_{low}, H_{low}), (T_{high}, H_{low}), (T_{low}, H_{high}) and (T_{high}, H_{high}). Then we can compare the co-effect of the temperature and humidity and their separate effects. More data are needed if light illumination is also considered.

Following the reviewer's requirement, we screened and checked all data from the database, but no such group of data are available because the current way for reporting device stability is brief and lack of standard. We found that most of the publications provide stability results under only one or two environmental conditions, which is what lots of current works do, and those with more than four stability results have gradient changes on only one environmental stressor or choose testing conditions randomly. That limits the validation of the co-dependency.

We have also tried to validate the co-dependency through the distribution of all data and choose temperature and humidity as an example. According to the Arrhenius model, the device performance decay rate, k, is a function of temperature, T.

$$k = A \cdot e^{\frac{-E_a}{k_B T}}$$

As the time to failure is inversely proportional to the degradation rate, T_{S80} is described as,

$$T_{S80} = B \cdot e^{\frac{E_a}{k_B T}}$$

Then,

$$\log(T_{S80}) = \frac{E_a}{k_B} \frac{1}{T} + C$$

E_a is the effective activation energy of the degradation process, k_B is the Boltzmann constant, and A, B, C are constants. As the equation shows, log(T_{S80}) is linear dependent with 1/T, and the slope of the line depends on the effective activation

energy, which represents the sensitivity to temperature and is usually constant in a specific process. If the co-dependency between temperature and humidity is involved, which means the co-effect of the high temperature and high humidity deviates greatly (larger or smaller) from the simple product of separate effects of stressors, the slope of the $\log(T_{S80})$ versus $1/T$ line will have a definite trend with the increase of humidity.

The $\log(T_{S80})$ versus $1000/T$ plots of devices at different humidity are shown in Figure R1 to R3. As the figures show, the slope of MAPbI_3 devices slightly increases from 2.2 to 2.7 below 60% RH and then drops a lot to -0.5. The negative slope at very large humidity may result from error caused by the small data size and selection bias. Devices with poor stability tend to be tested at lower testing temperature, while some stable devices tend to be tested under high environmental stresses (e.g., 85 °C, 85% RH), and this results in lots of high-lifetime datapoints in the double-85 area. For FAPbI_3 -based and all inorganic devices, no definite trend is observed, although negative slopes are also obtained at high humidity. Thus, specific mathematical relationships cannot be derived from the dataset. Moreover, most of the datapoints are far from the fitted lines, which means even if co-dependencies are considered in the indicator T_{S80m} , it may not lead to more precise conclusions, and more unknown parameters and uncertainty will be introduced instead.

Figure R1. The $\log(T_{S80})$ versus $1000/T$ plots of devices with medium tolerance factors ($0.85 < \alpha < 0.95$, mainly is MAPbI_3) at different humidity.

Figure R2. The $\text{Log}(T_{s80})$ versus $1000/T$ plots of devices with large tolerance factors ($\alpha > 0.95$, mainly is FAPbI_3 -based compositions) at different humidity.

Figure R3. The $\text{Log}(T_{s80})$ versus $1000/T$ plots of devices with small tolerance factors ($\alpha < 0.85$, mainly is all inorganic compositions) at different humidity.

Question 2:

In fig 2c, majority of the cells are said to have a T_{s80m} of many hundred hours. In practice, very few cells have such performance today. Thus T_{s80m} largely

overestimates stability, probably because it fails to consider how stressors reinforce each other.

Our idea of T_{S80m} comes from the accelerated degradation tests where degradation tests of hundreds of hours under harsh conditions are used to predict tens of years of the lifetime of devices. In such cases, the predicted lifetime is usually much larger than the test time. In this work, we use T_{S80m} , which estimates the lifetime under the reference conditions (27 °C, 0% RH and 1 sun illumination), as the indicator to uniformly assess device stability under many different testing conditions. Because a device will have a longer lifetime under milder conditions, T_{S80m} will have larger values than common testing results due to the milder reference conditions compared to the actual testing and working conditions. That is only conversion instead of overestimation.

In fact, we can choose the reference conditions freely. If we choose 85 °C, 85% RH and 1 sun illumination as the reference conditions, T_{S80m} values become much smaller, but that has no influence on the hypothesis test results. We have also added a detailed discussion and verification in the manuscript.

Revised manuscript:

“In addition, different reference conditions will not affect the conclusions. In accelerated degradation tests, hundreds of hours of tests under harsh conditions are used to predict tens of years of the device lifetime. T_{S80m} predicts the lifetime under the reference conditions (27 °C, 0% RH and 1 sun illumination), so the value of T_{S80m} is usually much larger than common testing results. We also choose 85 °C, 85% RH and 1 sun illumination as the reference conditions and recalculate T_{S80m} . The results show that all the data points only shift to smaller values without change in shape (Supplementary Fig. 21), and the hypothesis test conclusion about the tolerance factor remains the same (Supplementary Table 21).”

Supplementary Fig. 21 The distribution of all data with the reference conditions of 85 °C, 85% RH and 1 sun illumination. **a**, Histogram of $\log(T_{S80m})$ values for all data. **b**, The kernel density estimation of the $\log(T_{S80m})$ values for different tolerance factor

regions of 3D perovskite devices without encapsulation.

Supplementary Table 21. Statistical results of 3D perovskite devices with large, medium, and small tolerance factors with 85 °C, 85% RH and 1 sun illumination as the reference conditions.

Group	Sample	Average	Variance	Size	H ₀ accepted	μ	T _A /T _B ratio
1	Large (>0.95)	0.087	5.362	1691	Yes	0.619	1.858
	Medium (0.85~0.95)	-0.639	4.609	3608			
2	Small (<0.85)	0.838	8.097	402	Yes	1.282	3.605
	Medium (0.85~0.95)	-0.639	4.609	3608			

Question 3:

The range of values that gamma and activation energy can take is likely to introduce some uncertainty in the calculations. The authors should report what this uncertainty is.

The γ and E_a appear in the equations of acceleration factors of humidity and temperature respectively as parameters and have optional ranges according to some previous research in the field. In practice, the value of γ represents the sensitivity of the device lifetime to humidity, and E_a is that to temperature, (e.g., the acceleration factor of temperature will become larger under the same environmental conditions with a larger E_a value chosen). We have tried to change the values of E_a and γ to see their influence on the results. The value of E_a only influences the T_A/T_B ratios but does not change the conclusion even if some extreme values are taken, while γ has no influence on the hypothesis test results because of the linear relationship. We have added the results and the discussion in the manuscript.

Revised manuscript:

“For the range of parameters (E_a and γ), the different values will make T_{S80m} more sensitive or less to the environmental stresses. For example, with a larger E_a value, one device will achieve a higher T_{S80m} from $A_{temperature}$. Supplementary Table 19 and 20 show that the average of T_{S80m} is positively related to both E_a and γ . However, only E_a influences the hypothesis test results because of the exponential relationship, while the change of γ has the same effect on all the devices, which keeps the results the same. Thus, reasonable parameter values are needed for the lifetime estimation, but

the hypothesis test is less affected.”

Supplementary Table 19. Statistical results of doped organic HTL, inorganic HTL and HTL/Carbon devices with different E_a values.

E_a (eV)	Sample	Average	Variance	Size	H_0 accepted	μ	T_A/T_B ratio
0	Inorganic HTL	5.071	5.754	57	Yes	0.078	1.081
	Doped organic HTL	4.515	4.717	3074			
0	HTL/Carbon	6.642	4.324	94	Yes	1.753	5.774
	Doped organic HTL	4.515	4.717	3074			
0.6	Inorganic HTL	5.854	7.219	57	Yes	0.583	1.791
	Doped organic HTL	4.775	5.036	3074			
0.6	HTL/Carbon	7.132	5.745	94	Yes	1.969	7.163
	Doped organic HTL	4.775	5.036	3074			
1.2	Inorganic HTL	6.636	14.02	57	Yes	0.967	2.630
	Doped organic HTL	5.036	8.193	3074			
1.2	HTL/Carbon	7.621	11.15	94	Yes	2.090	8.084
	Doped organic HTL	5.036	8.193	3074			

Supplementary Table 20. Statistical results of doped organic HTL, inorganic HTL, and HTL/Carbon devices with different γ values.

γ	Sample	Average	Variance	Size	H_0 accepted	μ	T_A/T_B ratio
0.5	Inorganic HTL	5.161	7.219	57	Yes	0.583	1.791
	Doped organic HTL	4.082	5.036	3074			
0.5	HTL/Carbon	6.439	5.745	94	Yes	1.969	7.163
	Doped organic HTL	4.082	5.036	3074			
1	Inorganic HTL	5.854	7.219	57	Yes	0.583	1.791
	Doped organic HTL	4.775	5.036	3074			
1	HTL/Carbon	7.132	5.745	94	Yes	1.969	7.163
	Doped organic HTL	4.775	5.036	3074			
1.5	Inorganic HTL	6.259	7.219	57	Yes	0.583	1.791
	Doped organic HTL	5.181	5.036	3074			
1.5	HTL/Carbon	7.537	5.745	94	Yes	1.969	7.163
	Doped organic HTL	5.181	5.036	3074			

Question 4:

Furthermore, there is nothing new that the analysis reveals. Almost everything is well-known: all-inorganics are more stable than hybrid perovskites, n-i-p is more stable than p-i-n historically (now being overturned), 2D/3D heterostructures more stable than 3D... The idea that the tolerance factor strongly determines stability is perhaps shown here for the first time.

As we mentioned in the manuscript, one of the important meanings of our work is that

it demonstrates those well-known intuitions hold even if all available data is considered. The stability improvement strategies are all based on lessons learned from lots of controlled experiments. We are not repeating specific experiments to give results which is consistent with existing intuitions, but turning the well-known intuitions into definite and reliable conclusions with statistical analysis methods.

Moreover, besides qualitative conclusions, the statistical method also gives quantitative comparisons between the stability improvement capabilities of different strategies. For example, carbon-based devices have a 7 times longer lifetime but that of encapsulated devices is only 2.5. The macro assessment cannot be obtained from a single experiment, and the recorded values in the publications which usually focus on single strategies are not always applicable in all cases. That makes it difficult to choose a commercialization strategy combination, which we are trying to solve.

We also give some suggestions for choosing stable device structures according to our statistical results. Interestingly, there is still no device containing all those suggested options reported in the Perovskite Database, which makes an obvious suggestion for further experimental studies.

Reviewer 3:

This manuscript proposes big data driven perovskite solar cell stability analysis. The topic is interesting, and certainly consistent with the contents to be proposed to the readers of "Nature Communications". Moreover, the manuscript is well written and can be read with pleasure: this represents an important aspect in the current scenario of publications in international journals. Overall, I think that this manuscript has to be accepted, but the Authors should take into account the following minor revisions (in terms of bibliographic updates, grammar corrections and content deepening)

We thank the reviewer for the positive comments.

Question 1:

It is difficult to review a manuscript without page numbers!!!

We thank the reviewer for the suggestion. We have changed the layout and added page numbers in the manuscript.

Question 2:

Detailed revisions: I spent several hours reading this manuscript, and Authors are asked to follow carefully the attached PDF file where I highlighted some points to be addressed. The attached file also contains language mistakes and typos; some questions related to manuscript contents could also be present and Authors must consider them properly before submitting the revised manuscript.

We thank the reviewer for the help with the details. We have fixed all the errors marked in the PDF file.

Question 3:

The Introduction should give a wider overview on the present scenario related to hybrid photovoltaics, both in terms of recently published reviews and research articles. In particular, emerging sustainable, integrated and unconventional PV devices are missing and a paragraph on this topic is highly suggested to be added in the Introduction. Authors are invited to go through the literature published in the last six months on these issues, and also on concepts developed some years ago in this field. Some of them are also mentioned in the above mentioned PDF file.

We thank the reviewer for the suggestions. We have added some revisions in the main text.

Revised manuscript:

“The last decade has witnessed a rapid technological rush aimed at the development of emerging devices for solar energy conversion such as dye-sensitized cells¹, perovskite cells² and integrated devices³.”

“Thus, poly[bis(4-phenyl)(2,4,6-trimethylphenyl)amine] (PTAA),⁴⁵ poly(3-hexylthiophene-2,5-diyl) (P3HT)⁴⁶ and other new materials⁴⁷ are used to replace spiro-MeOTAD and dopants are removed to enhance stability.”

“Together with the advantages of low cost and simple preparation as previously reported⁵², carbon electrodes will be a promising candidate for commercialization.”

“Encapsulation has proved to be a simple and effective strategy to improve the external stability of PSCs by preventing the penetration of moisture and oxygen⁵⁵⁻⁵⁷ and to prevent lead leakage⁵⁸, which is a necessary part of commercialization.”

Question 4:

Authors should provide a clear explanation on the experimental error of the proposed research work. In particular, reproducibility of the phenomena described in the manuscript should be clearly stated in the “Results and Discussion” section; besides, some notes in the “Materials and Methods” section should be added highlighting which kind of experimental approach has been followed to check the reproducibility of the proposed system, the latter being of noteworthy importance in the present research field.

We thank the reviewer for the suggestions. We have discussed the uncertainty and reproducibility from four perspectives, namely the influence of the ranges of the parameters (E_a and γ), the influence of the reference conditions, the reliability of the hypothesis test method and the data from the Perovskite Database Project.

According to the results provided in the manuscript and SI, the value of E_a only influences the T_A/T_B ratios but does not change the conclusion even if some extreme values are taken, while γ has no influence on the hypothesis test results because of the linear relationship. The reference conditions only change the value of T_{S80m} because T_{S80m} represents the estimated lifetime under the reference conditions, but they have no influence on the distribution of the data and the hypothesis test results. For the hypothesis test method, some unaccepted hypotheses may be caused by small data sizes and large variances, but the strategies which show obvious stability improvement are credible. For the data from Perovskite Database, though only a small number of publications are included, the dataset is sufficient to draw clear and credible conclusions. We have added a discussion of error and reproducibility in the manuscript.

Revised manuscript:

“Discussion on uncertainty and reproducibility

The indicator T_{S80m} is calculated by converting three main environmental stresses, temperature, humidity and light intensity to separate acceleration factors and multiplying them with T_{S80} . Uncertainty will come from the co-dependencies between different stressors, the range of parameters (E_a in $A_{\text{temperature}}$ and γ in A_{humidity}) and the chosen reference condition.

For the range of parameters (E_a and γ), the different values will make T_{S80m} more sensitive or less to the environmental stresses. For example, with a larger E_a value, one device will achieve a higher T_{S80m} from $A_{\text{temperature}}$. Supplementary Table 19 and 20 show that the average of T_{S80m} is positively related to both E_a and γ . However, only E_a influences the hypothesis test results because of the exponential relationship, while the change of γ has the same effect on all the devices, which keeps the results the same. Thus, reasonable parameter values are needed for the lifetime estimation, but the hypothesis test is less affected.

In addition, different reference conditions will not affect the conclusions. T_{S80m} predicts the lifetime under the reference conditions (27 °C, 0% RH and 1 sun illumination), which is too mild compared to the actual testing and working conditions, thus the indicator seems to overestimate the device stability. We also choose 85 °C, 85% RH and 1 sun illumination as the reference conditions and recalculate T_{S80m} . The results show that all the data points only shift to smaller values without change in shape (Supplementary Fig. 21, SI), and the hypothesis test conclusion about the tolerance factor remains the same (Supplementary Table 21).

The detail of the hypothesis test method is described in Supplementary note 2 (SI). An accepted hypothesis, which means there is a statistically significant difference between two samples, requires large sample sizes, small variances and large average differences. Thus, the limitation of data (small data sizes and large variances) tends to give an unaccepted hypothesis, while the strategies which show obvious stability improvement are credible.

As mentioned above, the Perovskite Database contains stability data for 7419 devices with publication data from 2012.08.21 to 2021.05.21 at the time of writing. Note that only a small number of publications are included, but the dataset is sufficient to draw conclusions that are consistent with the current state of the field. However, the research focus of the PSC field changes over time (e.g., the change of mainstream perovskite compositions), so the conclusions are not always true and may be overturned in the future. Time-dependent statistical analysis is needed to draw dynamic conclusions, which is beyond the scope of this work.

”

“Materials and Methods

Data are downloaded from the Perovskite Database Project on 2022.01.18.

All the code for generating the analysis and the plots is written with Wolfram Mathematica, and available at <https://github.com/NK-ZZhang/PSC-stability.git> together with the final dataset file containing computed tolerance factors, T_{S80} , $A_{\text{temperature}}$, A_{humidity} , A_{light} and T_{S80m} values.

Hypothesis test method is used for the comparison between different strategies, and the details of the method are described in Supplementary note 2.”

Reviewer #1 (Remarks to the Author):

The authors have addressed all my concerns and the paper is ready for acceptance.

Reviewer #2 (Remarks to the Author):

I find the revisions made to the manuscript satisfactory. Publication is recommended.

Reviewer #3 (Remarks to the Author):

The manuscript has been properly amended and I recommend its publication.

Point-by-point reply to reviewer comments

We thank all the reviewers for their comments. Please find below the point-by-point reply to the comments, with the reply in blue color.

Reviewer 1:

The authors have addressed all my concerns and the paper is ready for acceptance.

We thank the reviewer for the positive recommendation towards acceptance.

Reviewer 2:

I find the revisions made to the manuscript satisfactory. Publication is recommended.

We thank the reviewer for the positive assessment for our revision.

Reviewer 3:

The manuscript has been properly amended and I recommend its publication.

We thank the reviewer for the positive recommendation towards publication.